# 1 Subglacial topography and ice flux along the English Coast of

# 2 Palmer Land, Antarctic Peninsula

Kate Winter[1], Emily A. Hill[1], G. Hilmar Gudmundsson[1], John Woodward[1]
[1]Department of Geography and Environmental Sciences, Faculty of Engineering and Environment, Northumbria
University, Newcastle upon Tyne, UK
*Correspondence to:* Kate Winter (k.winter@northumbria.ac.uk)
**Abstract.** Recent satellite data have revealed widespread grounding line retreat, glacier thinning, and associated
mass loss along the Bellingshausen Sea sector, leading to increased concern for the stability of this region of
Antarctica. While satellites have greatly improved our understanding of surface conditions, a lack of radio-echo
sounding (RES) data in this region has restricted our analysis of subglacial topography, ice thickness and ice flux.
In this paper we analyse 3,000 km of 150 MHz airborne RES data collected using the PASIN2 radar system (flown
at 3 – 5 km line spacing) to investigate the subglacial controls on ice flow near to the grounding lines of Ers,
Envisat, Cryosat, Grace, Sentinel, Lidke and Landsat ice streams as well as Hall and Nikitin glaciers. We find that
each outlet is topographically controlled, and when ice thickness is combined with surface velocity data from
MEaSUREs (Mouiginot et al., 2019), these outlets are found to discharge over $39.2 \pm 0.79$ Gt $a^{-1}$ of ice to floating
ice shelves and the Southern Ocean. Our RES measurements reveal that outlet flows are grounded more than 300
m below sea level, and that there is limited topographic support for inland grounding line re-stabilisation in a
future retreating scenario, with several ice stream beds dipping inland at ~5 degrees per km. These data reinforce
the importance of accurate bed topography to model and understand the controls on inland ice flow and grounding
line position as well as overall mass balance / sea level change estimates. RES data described in this paper are
available through the UK Polar Data Center: https://doi.org/10.5285/E07D62BF-D58C-4187-A019-
59BE998939CC (Corr and Robinson, 2020).

**Short summary**
Satellite measurements of the English Coast in the Antarctic Peninsula reveal that glaciers are thinning and losing
mass, but ice thickness data is required to assess these changes, in terms of ice flux, and sea level contribution.
Our ice penetrating radar measurements reveal that low-elevation subglacial channels control fast-flowing ice
streams, which release over 38 gigatons of ice per year to floating ice shelves. This topography could make ice
flows susceptible to future instability.



## 1 Introduction

Remote sensing satellites have increased our awareness and understanding of ice flows in Antarctica since their inception. In western Palmer Land, on the Antarctic Peninsula, Earth observation satellites have recorded widespread grounding line retreat (Christie et al., 2016; Konrad et al., 2018) and surface lowering (attributed to glacier thinning) in the last two decades (Wouters et al., 2015; Hogg et al., 2017; Smith et al., 2020), as well as surface velocity increases and significant mass loss (e.g. McMillan et al., 2014; Wouters et al., 2015; Martín-Español et al., 2016; Hogg et al., 2017), where ice flows contribute ~0.16 mm a$^{-1}$ to global mean sea level (Wouters et al., 2015).  Regional mass losses of  -56± 8 Gt per year between 2010 and 2014 (Wouters et al., 2015) exceed the magnitude of interannual variability predicted by surface mass balance models (van Wessem et al., 2014, 2016), suggesting that the English Coast of western Palmer Land is undergoing significant change. While satellites have greatly improved our understanding of surface conditions and changes across Antarctica in recent years, a lack of ice thickness and subglacial topographic measurements in western Palmer Land has restricted our analysis of the controls on ice flow, ice flux and grounding line stability along the English Coast (Minchew et al., 2018). As subglacial topography exerts a strong control over ice flow it is critical to collect and analyse ice penetrating radar (IPR) data close to the grounding line in understudied regions of Antarctica.

In this paper we present a new, freely available IPR dataset along the English Coast of western Palmer Land, where several outlet glaciers were named after Earth observation satellites in 2019, in deference to the critical role that satellites have played in measuring and monitoring the Antarctic Ice Sheet (Fig. 1). We combine this new geophysical dataset with satellite measurements of ice flow speeds from MEaSUREs (Mouginot et al., 2019) to provide an improved picture of the subglacial controls on ice flows draining the English Coast, and directly assess the improvements to our understanding of bed topography and ice flux in the region as a result of such high-resolution IPR datasets.

## 2 Location and previous work

The English Coast of western Palmer Land contains numerous outlet glaciers which flow at speeds of ~0.5 to 2.5 m per day (Mouginot et al., 2019), from accumulation areas in central Palmer Land, towards ice shelves in the Bellingshausen Sea sector of Antarctica (Fig.1). A map of surface ice flow speeds in Figure 1 shows how the recently named Ers, Envisat, Cryosat and Grace ice streams drain into the fast-flowing George VI Ice Shelf, where floating ice connects Palmer Land to Alexander Island. Further south, Sentinel Ice Stream passes the local grounding line to form a floating tongue, connected in part to the neighbouring George VI Ice Shelf. Moving south of George VI Ice Shelf, Hall Glacier, Nikitin Glacier and Lidke Ice Stream each flow into Stange Ice Shelf. Whilst these outlet flows have separate accumulation zones that border the large Evans Ice Stream catchment (which drains into the Weddell Sea, on the other side of the Antarctic Peninsula) (Fig.1), their distinct flow units converge along the English Coast, at the local grounding zone. At the southern extremity of the English Coast, Landsat Ice Stream flows close to the catchment-defined boundary between the Antarctic Peninsula and West Antarctica. Slow flowing, almost stagnant ice separates the two tributary flows of Landsat Ice Stream for much of its length (Mouginot et al., 2019).



Our understanding of the English Coast of western Palmer Land is driven by data accessibility. Fast ice flow, and
heavily crevassed surfaces have largely restricted in-situ data collection in this region. The first Antarctic-wide
ice thickness and subglacial topography datasets: Bedmap (Lythe et al., 2001) and Bedmap2 (Fretwell et al., 2013)
relied on sparse IPR measurements for interpolation in this region of Antarctica. As a result, there are large
uncertainties in bed topography and ice thickness along the English Coast, which limit our understanding of
regional ice dynamics (Minchew et al., 2018). Inaccurate ice thickness and bed topography also hinder our ability
to assess the sensitivity of this region to future change using numerical ice flow models. Previous work has
therefore made use of more readily available satellite data, such as optical images, altimeter data and synthetic
aperture radar (SAR) measurements to assess regional change. Numerous studies have used these detailed datasets
to report on, and model recent changes in surface elevation and ice flow along the Bellingshausen Coast (e.g.
Pritchard et al., 2012; Christie et al., 2016; Hogg et al., 2017; Minchew et al., 2018), as well as Antarctica as a
whole (e.g. Helm et al., 2014; McMillan et al., 2014; Konrad et al., 2018; Smith et al., 2020). Collectively, this
work has highlighted a number of potential vulnerabilities in western Palmer Land. Recent mass loss of George
VI Ice Shelf and Stange Ice Shelf (totalling an estimated 11 Gt a$^{-1}$) (Rignot et al., 2019) raised concern that English
Coast outlet glaciers could be susceptible to the marine ice sheet instability mechanism (Wouters et al., 2015) -
where grounding-lines have a tendency to accelerate down a retrograde slope in the absence of compensating
forces (like buttressing ice shelves) (Schoof, 2007; Gudmundsson et al., 2012). These concerns are compounded
by recent changes in the grounded ice flows along the English Coast. Wouters et al. (2015) reported an average
surface lowering of ~ 0.5 m a$^1$ along the coastline between 2010-2014, whilst Hogg et al. (2017) calculated a 13%
increase in outlet glacier ice flow between 1993 and 2015. Importantly, if surface thinning and ice flow
acceleration across western Palmer Land continue in the future, dynamical imbalance could lead to further draw
down of the interior ice sheet (like it has done in other areas of Antarctica, (e.g. Shepherd et al., (2002); Rignot
(2008); Konrad et al. (2018)), leading to increased ice discharge into the ocean (Gudmundsson, 2013; Wouters et
al., 2015; Fürst et al., 2016; Kowal et al., 2016; Minchew et al., 2018), with resultant sea level rise. New, high-
resolution measurements of ice thickness and subglacial topography close to the grounding line will improve our
understanding of ice dynamics along the English Coast, and enable more accurate modelling of current conditions,
and forward-looking estimations.
**3 Methods**
Data sets outlined in subsections 3.1 – 3.3 are freely available to download. Download links are provided in Sect.7.
**3.1 Airborne radio echo sounding acquisition, processing, and visualisation**
In the austral summer of 2016/2017, the British Antarctic Survey Polarimetric-radar Airborne Science Instrument
(PASIN2) ice sounding radar system was used to acquire ~3,000 line km of radio-echo sounding (RES) data along
the English Coast of western Palmer Land, at ~3 – 5 km line spacing (Corr and Robinson, 2020). PASIN2 operates
at a frequency of 150 MHz, using a pulse-coded waveform at an effective acquisition rate of 312.5 Hz and a
bandwidth of 13 MHz. Technical details of the RES system are available in Jeofry et al. (2018) and references
therein. Differential GPS was used to record aircraft position (with an accuracy better than ± 1 m) and RES data
were collected at an average flying velocity of 55 m s$^{-1}$. Along-track processing of the data results in an output
data rate of 5 Hz, which produces an average spacing between radar traces of 11 m.






For the processing of the data, a coherent moving-average filter, commonly referred to as an unfocused SAR, was
used on the range compressed data. The onset of the bed reflector was first automatically picked using first-break
picker of the ProMAX seismic processing software with all picks then checked afterwards and corrected by hand
if necessary. The delay time of the bed reflector picks were covered to range using a standard electromagnetic
wave propagation speed in ice of 0.168 m ns$^{-1}$ and a correction of 10 m to account for the near-surface high-
velocity firn layer (Dowdeswell and Evans 2004; Vaughan et al., 2006). Ice thickness was calculated by
subtracting surface elevation measurements (derived from radar/laser altimeters for aircraft terrain clearance) from
bed reflector depth picks. Internal crossover analysis (measurements of ice thickness at the same position) yield a
standard deviation of 13 m at line intersections, with no systematic line-to line biases. Independent crossover
analysis, with NASA's airborne Operation Ice Bridge (OIB) radar data (collected from November 2010 –
November 2016), yields a higher standard deviation of 75 m. Whilst this value is similar to recent independent
crossover analysis in other surveys, like the Pensacola-Pole basin in East Antarctica (where Paxman et al. (2019)
reported a standard deviation of 57.7 m), we note that our standard deviation is skewed by a number of high
elevation OIB flights and a relatively small number of high crossover misfits over steep subglacial topography,
where the bed elevation is more difficult to determine. As such, we use the internal crossover analysis value of 13
m for our RES errors.

RES transects were visualised in 2D in Reflexw radar processing software (version 7.2.2; Sandmeier Scientific
Software) where an energy decay gain was applied to compensate for geometric spreading losses in the radargram
(Daniels et al., 2004). Opendtect seismic interpretation software (2015) was employed to plot radargrams in real
space using DGPS co-ordinates, to enable three-dimensional analysis of RES data.

**3.2 Mapping subglacial topography and ice thickness**
Airborne RES data presented in this paper have been incorporated in the new BedMachine dataset; a self-
consistent dataset of the Antarctic Ice Sheet based on conservation of mass, which has a resolution of 500 m
(Morlighem, 2019; Morlighem et al., 2019). As a result, data presented in this paper has already been combined
with numerous other RES survey data (including OIB data) to produce continent-wide ice thickness and subglacial
topography maps (Fig. 2b). Whilst Morlighem et al. (2019) report potential vertical errors of ~100 m in central
Palmer Land, these values decrease towards the coast, where RES measurements are more frequent (Morlighem,
141 2019).


**3.3 Surface flow speeds**
Surface flow speeds are extracted from MEaSUREs phase-based Antarctica ice velocity map which has a
resolution of 450 m (Mouginot et al., 2019) (Fig. 1a). This data set combines interferometric phases from multiple
satellite interferometric synthetic-aperture radar systems, with additional data, including tracking-derived velocity
to maximise coverage from 1996 to 2018. Across western Palmer Land the average flow speed error is estimated
to be > 4 m a$^{-1}$.

**3.4 Calculating ice flux**



Using surface flow speeds (Mouginot et al., 2019), and the high-resolution ice thickness measurements along our RES transects, we calculate ice flux across fixed gates delineated for each of the named ice streams and glaciers along the English Coast (Fig. 2). These flux gates are delineated along RES transects close to the grounding line and they span the width of each outlet. Ice flux ($q$) for each ice stream or glacier ($j$) is calculated following Eq. (1):

$$q_j = \sum_{i=1}^{n} h_{ij} w_{ij} \vec{v}_{ij} \rho \qquad (1)$$

where $i$ is an equally spaced bin along the length of the flux gate, $w$ is the bin width (which is fixed to 1 m for all outlets, and is sufficiently small that the solution is not sensitive to a bin width smaller than this), $\vec{v}$ is the velocity normal to the flux gate, and $\rho = 916.7 \text{ kg m}^{-3}$ is ice density. For simplicity, we are assuming that surface velocities and ice density are constant with depth. To examine the impact of incorporating high resolution RES data into gridded bed topography datasets we directly compare ice flux from Bedmap2 (Fretwell et al., 2013) (which has a resolution of 1 km) with the radar picks described in Section 3.1, which are included in BedMachine (Morlighem, 2019) (Figure 2c). For these calculations we use the same flux gates, phase-based ice velocities and ice density; simply replacing RES ice thickness for Bedmap2 ice thickness. Using available errors in velocity and ice thickness datasets we calculate errors in our calculated ice flux ($\sigma_q$) for each glacier following Eq. (2):

$$\sigma_q = \sqrt{\sigma_v^2 + \sigma_h^2} \qquad (2)$$

where $\sigma_v = \sum_{i=1}^{n} h_{ij} w_{ij} d\vec{v}_{ij} \rho$ and $\sigma_h = \sum_{i=1}^{n} dh_{ij} w_{ij} \vec{v}_{ij} \rho$ are the contribution of errors in velocity ($dv$) and ice thickness ($dh$) to the errors in ice flux respectively. Ice flux and associated error bars for each outlet are shown in Figure 2c.

## 4 Results

Our airborne RES transects map subglacial topography and ice thickness down the English Coast, from Ers Ice Stream to Landsat Ice Stream. Whilst our results and discussion focus on the largest outlets, ice flux from each of the named outlets is presented in Figure 2. The complete RES dataset (marked in Fig. 1) is freely available to download from the UK Polar Data Centre (see Sect.7 for more details).

### 4.1 Ers Ice Stream

Close to the grounding line, Ers Ice Stream reaches a maximum flow speed of just over 940 m a$^{-1}$ (averaging out at ~2.5 m per day) (Mouginot et al., 2019). This ice originates from central Palmer Land (Fig. 1), where ice flows across the west of the Antarctic Peninsula, towards Ers Ice Stream. In the upper catchment, flow speeds of ~400 m a$^{-1}$ (Mouginot et al., 2019) are recorded along RES transect Ers 6 (Fig 3a). A succession of airborne RES transects in Figure 3c show how this fast-flowing ice is channelised towards the coast, through a subglacial depression ~8 – 14 km wide. As ice flows through this channel, towards the local grounding line (marked in white on Fig. 3a), ice thickness reduces from a maximum of ~1400 m (along transect Ers 6) to between 580 and 610 m (along transect Ers 1), where the ice flow is grounded ~400 m below sea level. Ice flux calculated along this radar transect suggests that Ers Ice Stream contributes over 7.2 ± 0.15 Gt a$^{-1}$ to George VI Ice Shelf (Fig. 2c). Although



this flux gate represents the main trunk of Ers Ice Stream (Fig 2a, 2b), neighbouring ice flow from the lateral
margins of the ice stream (where ice flows at ~210 – 390 m a$^{-1}$) will, of course, add to this value.

**4.2 Cryosat Ice Stream**
A central flow unit, more than 14 km wide, distinguishes Cryosat Ice Stream from neighbouring regions of slower
flowing ice along the English Coast (Fig. 1). Whilst surrounding ice flows at ~100 m a$^{-1}$, flow speeds in the ice
stream range from 400-500 m a$^{-1}$ inland (along RES transect Cryosat 3), to 950 m a$^{-1}$ (~2.6 m per day) (Mouginot
et al., 2019) along RES transect Cryosat 1 - which was traversed close to the Antarctic Surface Accumulation and
Ice Discharge (ASAID) grounding line (Bindschadler et al., 2011) (Fig. 4a). Figure 4a shows how the main flow
of Cryosat Ice Stream is joined by a smaller tributary to the south, where ice flow speeds increase from 180 m a$^{-}$
$^{1}$ along transect Cryosat 4 to over 400 m a$^{-1}$ (Mouginot et al., 2019) along transect Cryosat 1 (traversed ~5 km
from the ASAID grounding line). In both flow units, the subglacial bed remains well below sea level along the
length of each transect. Close the grounding line, along transect Cryosat 1, the glacial bed is between 450 and 800
m below sea level, where overlying ice is 500 – 900 m thick. Although subglacial topographic depressions are
visible in-land (where subglacial peaks which reach over 500 m from the bed help to define the low elevation
topography), subglacial troughs become more defined towards the coast, where ice is guided through several
almost U-shaped troughs (Fig. 4b). This is most obvious in RES transect Cryosat 1, where the main flow of ice is
channelled through a ~14 km wide, 300 m deep subglacial trough close to the local grounding line, whilst the
smaller (southern) tributary flow is directed through a ~400 m deep trough, which is ~3 km wide at its base (Fig.
4b). The flow units of Cryosat Ice Stream collectively discharge just under 6 ± 0.14 Gt a$^{-1}$ of ice across the
grounding line. The ice flux profile in Fig. 4b shows how much of this flux is discharged through the deep and
fast flowing central sector of the ice stream, rather than the deeper southern tributary.

**4.3 Sentinel Ice Stream**
In the MEaSUREs velocity map, Sentinel Ice Stream appears to have the widest outflow of the English Coast,
reaching a width of over 20 km. The main trunk of the ice stream curves round from an almost southerly flow
direction, to a more westerly direction along its length (Fig. 4c) as ice flow speeds increase from ~350 m a$^{-1}$ (along
RES transect Sentinel 5) to ~800 m a$^{-1}$ (closer to the grounding zone, along transect Sentinel 1) (Mouginot et al.,
2019). Whilst the subglacial bed remains well below sea level in all transects (at elevations in the region of -500
to -680 m), fluctuations in subglacial topography and ice thickness are recorded along and down flow in successive
RES transects (Fig. 4d). The largely unconfined ice flow in transect Sentinel 5 becomes more confined down flow
due to the emergence of higher elevation subglacial topography along the lateral margins of Sentinel Ice Stream.
These subsurface conditions are concurrent with ice thickness measurements (where maximum ice thickness
decreases down flow, from ~1200 m in transect Sentinel 5 to ~550 m in Sentinel 1), as well as surface velocity
measurements, which reveal increasing flow speeds in the central trunk of Sentinel Ice Stream with distance down
flow. The total flux of Sentinel Ice Stream is 6 ± 0.14 Gt a$^{-1}$. Whilst this flux will be added to by flow from the
south (where enhanced flow speeds are recorded, but they are ~3 x slower than the central trunk of Sentinel Ice
Stream), there will be much less flux in the north, where ice flows at a few 10s of metres per year (Mouginot et
al., 2019) (Fig. 4c), over higher elevation subglacial topography (~400 m higher than the base of the subglacial
trough).





### 4.4 Hall Glacier

Hall Glacier is the most northern tributary flow of the Stange Ice Shelf (Fig. 1). Surface flow speeds increase from RES transect HNL 6 (close to the onset of streaming flow) - where ice flows just over 100 m a$^{-1}$, to RES transect HNL 1 (~1.5 – 9.5 km from the ASAID grounding line and 14 km from HNL 6), where ice flow speeds reach 380 m a$^{-1}$ (Mouginot et al., 2019) (Fig. 5a). These enhanced flow speeds clearly differentiate Hall Glacier from the almost stagnant neighbouring ice flow (<10 m a$^{-1}$) along its lateral margins in Figure 5a. This figure shows how the fast-flowing portion of the outlet glacier decreases in width from ~15 km inland to ~8 km along RES transect HNL 2. This reduction in width coincides with a change in subsurface topography and ice thickness (Fig. 5c). Whilst a shallow subglacial depression is apparent upstream, in RES transect HNL 5 (where the subglacial bed is ~500 m below sea level and ice thickness reaches a maximum of 750 m), a much deeper channel is recorded down flow, where ice up to 930 m thick is channelised through high elevation subglacial topography. The profile in Figure 5b (derived from the flux gate marked on Figure 5a) shows the impact this subglacial topography and ice thickness has on ice flux. Flux is greatest along the central trunk of Hall Glacier where a ~7 km wide subglacial channel supports ice flow speeds of more than 350 m a$^{-1}$ (Mouginot et al., 2019). Over the whole flux gate, Hall Glacier contributes ~1.8 ± 0.04 Gt a$^{-1}$ of ice to the Stange Ice Shelf, which drains into the Bellingshausen Sea sector of the Southern Ocean.

### 4.5 Nikitin Glacier

Situated between Hall Glacier and Lidke Ice Stream, Nikitin Glacier maintains flow speeds in the region of 200-450 m a$^{-1}$ (Mouginot et al., 2019), as ice flow from central Palmer Land begins to stream towards the Stange Ice Shelf (Fig. 5). For much of its length, Nikitin Glacier flows through a 15 km wide subglacial channel, where ice thicknesses up to 1000 m flow over a glacial bed situated well below sea level (with elevations of -400 to -700 m). This low-elevation subglacial topography combined with thick ice flows and enhanced ice flow speeds enable Nikitin Ice Stream to contribute over 2.13 ± 0.05 Gt a$^{-1}$ of ice to the Stange Ice Shelf. Whilst it is difficult to precisely define the point at which this ice begins to float in our radargrams, it is worth noting that complex and highly reflective RES returns beneath Nikitin Ice Stream in transect HNL 1 suggest that the ice stream could be afloat here. This finding is coincident with the positioning of the ASAID grounding line (Bindschadler et al., 2011) (marked as a white line in Figure 5a), which is derived from satellite data.

### 4.6 Lidke Ice Stream

The MEaSUREs dataset (Mouiginot et al., 2019) presented in Figure 5a, shows how Lidke Ice Stream is fed by two tributary flows which coalesce close to RES transect HNL 4, where ice begins to flow along a central trunk at flow speeds in the region of 350 – 420 m a$^{-1}$ (Fig. 5a). Although Lidke Ice Stream is linked to neighbouring Nikitin Ice Stream in its upper catchment, a clear separation between the two ice streams is recorded down flow, where the enhanced flow units become separated by a region of almost stagnant ice (< 10 m a$^{-1}$). RES transects in Figure 5c show how this slow-moving ice sits on top of relatively high elevation subglacial topography (with elevations of -380 to -500 m). This raised topography helps to define the northern margin of Lidke Ice Stream, which flows through much lower elevation subglacial topography, situated ~600 – 800 m below sea level.



In RES transects HNL 4 and HNL 5 (traversed close to the onset of streaming flow) numerous peaks and troughs
dominate the subglacial topography returns, resulting in spatially variable ice thickness and ice flux. However,
further down flow, and closer to the grounding line, subglacial topography is more subdued, with the emergence
of a depressed subglacial channel (reaching a maximum depth of 810 m below sea level), where ice up to 1250 m
thick achieves surface flow speeds in the region of 400 m $a^{-1}$ (Mouginot et al., 2019) at the grounding zone. In a
flux gate along HNL3 (marked in Fig. 5a), Lidke Ice Stream is calculated to contribute >2.7 ± 0.01 Gt $a^{-1}$ to the
Stange Ice Shelf. The flux profile in Figure 5b shows how this value is distributed across the glacier – with high
flux values recorded in areas which have low elevation subglacial topography, thicker ice, and fast ice flow.

**4.7 Landsat Ice Stream**
Landsat Ice Stream (situated close to the catchment-defined boundary between the Antarctic Peninsula and West
Antarctica) is formed of a northern and southern tributary, with ice flow converging at, or close to the ASAID
grounding line (Fig. 6). Both tributaries have similar characteristics: they each reach flow speeds in excess of 500
m $a^{-1}$ in the centre of the ice flow (along RES transect Landsat 3) before flow begins to accelerate downstream (to
over 700 m $a^{-1}$ near transect Landsat 1) (Mouginot et al., 2019). Between the two tributaries, flow speeds are much
lower, ranging from 40 m $a^{-1}$ (25 km inland of the grounding line), to ~100 m $a^{-1}$ (along RES transect Landsat 1,
traversed close to the local grounding line) (Mouginot et al., 2019). A sequence of airborne RES transects in
Figure 6c show that these flow speeds reflect subglacial topography. Both tributaries flow through deep subglacial
basins (situated ~700 m below sea level), where ice flows up to 900 m thick are increasingly channelised towards
the coast by higher subglacial topography along the ice stream's lateral margins. Along RES transect Landsat 2,
ice flux gates across the north and south tributary flows combine to produce a total ice flux of 7.2 ± 0.13 Gt $a^{-1}$.
Between these two flow units ice flux is substantially lower, because of lower surface flow speeds, elevated
subglacial topography, and reduced ice thickness.

**5 Discussion**
English Coast ice streams and glaciers contribute over 39.2 ± 0.79 Gt $a^{-1}$ of ice to floating ice shelves in the
Bellingshausen Sea. This ice flows from the center of Palmer Land, towards the coast, where discrete ice flows
develop in line with, and as a result of depressed subglacial topography - in a region of Antarctica where the
glacial bed is situated well below sea level. In the following paragraphs, we briefly discuss the main features of
each major ice stream (documented in the results) from north to south. The significance of the radar data set is
presented in Sect. 6.

**5.1 Ers Ice Stream**
Ers Ice Stream, at the northern extremity of our study site, produces the largest ice flux of all English Coast ice
streams (Fig. 2c). This is the result of elevated surface flow speeds (Mouiginot et al., 2019), substantial ice
thicknesses and pronounced subglacial topography, which, for the most part, channelises ice through a wide
subglacial depression (Fig. 3c). Enhanced ice flow is also recorded on either side of the subglacial channel, where
surface flow speeds >100 m $a^{-1}$ (Mouiginot et al., 2019) contribute over ~100,000 tonnes of ice to George VI Ice
Shelf per year. This enhanced ice flow makes it difficult to precisely map the lateral margins of the ice stream and
fully assess the individual contribution of Ers Ice Stream to English Coast ice flux. However, it is clear that this



area of the English Coast contributes substantial and continued ice flux to George VI Ice Shelf, as a result of high
surface flow speeds, thick ice and deep subglacial topography.

**5.2 Cryosat Ice Stream**
Although ice flux from Cryosat Ice Stream is >50% lower than neighbouring Ers Ice Stream, it boasts the greatest
surface flow speeds of the English Coast: flowing at a maximum of 950 m a$^{-1}$ (Mouiginot et al., 2019) (averaging
out at >2.6 m per day). These enhanced ice flow speeds are recorded along the width of the ice stream, where
thick ice flows through, and over multiple, deep incisions in the basal topography (Fig. 4b). Fig. 4a shows how
these ice flow speeds are maintained across the grounding zone, as ice flows into George VI Ice Shelf. As the ice
shelf buttresses the inland ice flow of Cryosat Ice Stream, further thinning of the ice shelf could reduce resistive
stress (buttressing) at the grounding line, subsequently increasing ice discharge in this region (Tsai et al., 2015;
Minchew et al., 2018).

**5.3 Sentinel Ice Stream**
Pronounced topographic depressions in most of the cross-flow radar lines that transect Sentinel Ice Stream (Fig.
4d) suggest a degree of topographic confinement for Sentinel Ice Stream, which is grounded >500 m below sea
level. Whilst this confinement helps to channelise over 6 Gt a$^{-1}$ of ice towards the local grounding line currently,
along-flow radargrams in Figure 7a show how the ice stream might respond to future ingress of the grounding
line position (e.g. Christie et al., 2016). Ice stream thickness fluctuates in conjunction with subglacial topography
down the main trunk of the ice stream - from the upper catchment of the ice stream to the floating ice tongue,
which is recorded by bright, white RES reflectors in Fig, 7a. These bright reflectors help to highlight the grounding
zone (MacGregor et al., 2011), where ice flexes in response to tidal modulation (e.g. Rosier and Gudmundsson,
2018). Annotations in Figure 7a point out a range of previously unknown subglacial features beneath Sentinel Ice
Stream, like reverse subglacial slopes close to the grounding zone (which decline inland at ~5.5 – 4.5 ° per km),
as well as more raised topographic features further inland. These measurements are critical for simulations of
groundling line retreat. They show that a retreat of the grounding line into deeper water could allow thicker ice to
reach floatation, which would increase glacier driving stress and ice flux across the grounding line (Tsai et al.,
2015), with immediate implications for ice flow speed, ice discharge, and meltwater contribution to the Southern
Ocean (Minchew et al., 2018). RES measurements inland of the present-day grounding line reveal a steep reverse
bed-slope, which after an initial retreat of the grounding line (due to some forcing) could promote unstable
(runaway) grounding retreat (e.g. Schoof 2007; Jamieson et al., 2012; Kleman and Applegate, 2014). However,
elevated subglacial topography ~10 km inland of the current grounding line could potentially act as a pinning
point for future ice stream re-grounding (Favier et al., 2016) (Fig, 7a). Our RES measurements will allow these
potential instabilities to be explored in new, high-resolution numerical modelling simulations.

**5.4 Hall Glacier, Nikitin Glacier and Lidke Ice Stream**
Further down the English Coast, Hall Glacier, Nikitin Glacier and Lidke Ice Stream are clearly discernible in
maps of surface ice flow speeds (Mouiginot et al., 2019) (Fig. 1) and subsurface topography maps, like Bedmap2
(Fretwell et al., 2013) and the newer, higher resolution BedMachine (Morlighem et al., 2019) (Fig. 2). These maps
show how discrete ice flow units develop in accordance with subglacial depressions, where elevated subglacial
topography between tributaries help to promote independent, channelised ice flow towards the coast (Fig. 5). All
three ice flows converge in the floating Stange Ice Shelf, where they release a combined ice flux of ~6.7 Gt a$^{-1}$.
The zone between grounded and floating ice is discernible in satellite data (Bindschadler et al., 2011) (noted by
the ASAID grounding line in Figure 5a) and in our RES dataset, where bright subglacial reflections suggest water
ingress (MacGregor et al., 2011) along line HNL 1 (Fig.5). These independent data sets mark the same grounding
zone position along the English Coast. Whilst our radargrams do not extend seaward of transect HNL 1, we
hypothesise that the 8 km digression of the ASAID grounding line in Figure 5a could reflect the subglacial
extension of the deep subglacial trough beneath Hall Glacier. This relative extension of the grounding line shows
the impact subglacial troughs can have on grounding line location and potentially grounding line stability (as
noted in other regions of Antarctica, by Jamieson et al. (2012)). Should the grounding line migrate in the future,
relatively small-scale subsurface features like these could result in substantially different reactions from
neighbouring ice flows, like Hall Glacier, Nikitin Glacier and Lidke Ice Stream.
**5.5 Landsat Ice Stream**
The final radar transects in our survey were flown across Landsat Ice Stream (Fig. 6). These radargrams reveal
topographically confined ice flow along two discrete tributaries (north and south) for more than 15 km. These ice
streams, which flow at speeds >500 m a$^{-1}$ (Mouiginot et al., 2019) contribute over 7.2 Gt a$^{-1}$ of ice to the
Bellingshausen Sea. Along-flow lines presented in Figure 7b show the differences in ice thickness and subglacial
topography between the north and south tributaries of Landsat Ice Stream, which are each grounded >700 m below
sea level. The north tributary flows across a remarkably flat bed for most of its length, but this is punctuated by a
region of elevated subglacial topography ~5 km inland of the current grounding line, which is ~100 m higher than
surrounding bed returns (Landsat 6 transect, Fig. 7b). Whilst this generally flat, low-elevation subglacial bed
could enable rapid grounding line retreat in response to mass balance changes and/or applied oceanic forcings
(Weertman, 1974; Jamieson et al., 2012), this region of elevated subglacial topography could act as a temporary
pinning point for re-grounding in a retreating ice sheet scenario. A similar potential pinning point is located much
further inland of the grounding zone on the south tributary of Landsat Glacier (RES transect Landsat 7, Fig. 7b).
Here, flat subglacial topography (situated ~600 m below sea level) extends ~12 km inland of the current grounding
line, until bed topography lowers slightly and then inclines by 120 m over 2 km. Beyond this point, there is a
reverse slope, dipping inland at 3.5° per km. This subglacial topography correlates with satellite-derived surface
ice flow speeds (recorded by Mouginot et al., 2019): enhanced flow is recorded along RES transect Landsat 1,
where bright subglacial reflectors suggest the presence of subglacial water (MacGregor et al., 2011). These
reflections, which extend inland of the ASAID grounding line could provide the subglacial evidence to corroborate
recent satellite-derived measurements of inland grounding-line migration in this region of Antarctica (Christie et
al., 2016; Konrad et al., 2018). As warm circumpolar deep water resides at ~300 m depth in the neighbouring
ocean (Kimura et al, 2015) any relatively warm water ingress inland could promote ice dynamical imbalance in
this region of Antarctica and lead to further drawdown of ice from the interior (as reported by Hogg et al., 2017).
**6 Significance of the dataset**
Our RES data set provides the scientific community with over 3,000 km of airborne RES data along the English
Coast of the Antarctic Peninsula. The density of transects (at 3 – 5 km line spacing), and coverage so close to the



grounding line is unusual. Resultant latitude, longitude and elevation data (available from the Polar Data Centre)
adds considerable ice thickness and subglacial topographic information to this area of Antarctica, where pre-
existing and reliable ice penetrating radar data sets are more infrequent than other regions of the continent (like
central Graham Land or Pine Island Glacier). Figure 2c shows the impact this dataset has on ice flux estimations,
compared to older continent-wide compilations of ice thickness and subglacial topography. Ice flux calculated
using our new RES measurements yields a total ice flux $39.2 \pm 0.79$ Gt a$^{-1}$. Whist this is similar to flux derived
from Bedmap2 ice thickness measurements (Fretwell et al., 2013) ($39.8 \pm 7.1$ Gt a$^{-1}$), high errors are associated
with this older dataset, and there are regional discrepancies between the ice flux measurements. Along the upper
stretch of the English Coast (Ers, Envisat, Cryosat, Grace and Sentinel ice streams, and Hall Glacier), Bedmap2
overestimates ice flux by ~2.6 Gt a$^{-1}$ and along the southern outflows (Nikitin Glacier, Lidke Ice Stream and
Landsat Ice Stream) Bedmap2 underestimates ice flux by ~1.3 Gt a$^{-1}$, compared to our RES ice thickness
measurements. Due to the coarse resolution and limited number of IPR measurements incorporated in Bedmap2,
errors in ice thickness are on the order of 100s of metres and range from 13-45% of total ice thickness across our
flux gates. RES data presented here are of much higher resolution, where calculated internal errors in ice thickness
are < 3% of ice thickness measurements along each flux gate. Inclusion of these new IPR measurements in
BedMachine (Morlighem, 2019) has greatly improved the resolution and accuracy of the latest continent-wide
subglacial topography and ice thickness maps (Figure 2). Accurate, high resolution ice thickness and subglacial
bed measurements are crucial for understanding ice flow and modelling ice dynamics. It must therefore remain a
future research priority to collect more IPR data across the Antarctic Ice Sheet, and target regions that remain
geophysically understudied. These IPR measurements should be collected along- and across-flow to capture
small-scale topographic perturbations in the subglacial bed (e.g. Figure 7), which are critical for assessing the
potential for grounding line retreat and marine ice sheet instability.

**7 Data Availability**

Radio echo sounding data used in this paper, from Corr and Robinson (2020) are available through the UK Polar
Data Center: https://doi.org/10.5285/E07D62BF-D58C-4187-A019-59BE998939CC. Data related to surface ice
velocity from MEaSUREs (Mouginot et al., 2019) can be downloaded here: https://nsidc.org/data/NSIDC-
0754/versions/1. Maps of subglacial topography and ice thickness can be accessed from the BedMachine
repository (Morlighem, 2019): https://nsidc.org/data/nsidc-0756.

**8 Conclusions**

Ice penetrating radar transects along the English Coast of western Palmer Land in the Bellingshausen Sea sector
of the Antarctic Peninsula reveal multiple topographically confined ice flows, grounded ~300 – 800 m below sea
level. New ice thickness data, combined with satellite derived surface flow speeds from MEaSUREs (Mouiginot
et al., 2019) allow us to improve ice flux calculations along the recently named Ers, Envisat, Cryosat, Grace,
Sentinel and Landsat ice streams as well as the previously titled Hall and Nikitin glaciers and Lidke Ice Stream.
At a time when satellites are recording widespread grounding line retreat (Christie et al. 2016; Konrad et al.,
2018), surface lowering (attributed to glacier thinning) (Wouters et al., 2015; Hogg et al., 2017; Smith et al., 2020)
and significant mass loss (McMillan et al., 2014; Wouters et al., 2015; Martín-Español et al., 2016; Hogg et al.,
2017) along the English Coast, our radio-echo-sounding (RES) dataset provides the high resolution ice thickness,



and subglacial topography data required for change detection. These measurements and analysis will improve
simulations of Antarctic coastal change and associated global sea level estimations.

**Author contributions**
All authors contributed to the writing and editing of the paper. K. Winter was the principle investigator of the
project, which was instigated by G. H. Gudmundsson and guided by J. Woodward. Ice flux calculations were
provided by E. A. Hill.

**Competing Interests**
The authors declare that they have no conflict of interest.

**Acknowledgements**
RES data were collected by the British Antarctic Survey aerogeophysical group in the austral summer of
2016/2017 and data were pre-processed by Hugh F. J. Corr (British Antarctic Survey). We thank all those involved
in the process of planning and collecting data.

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

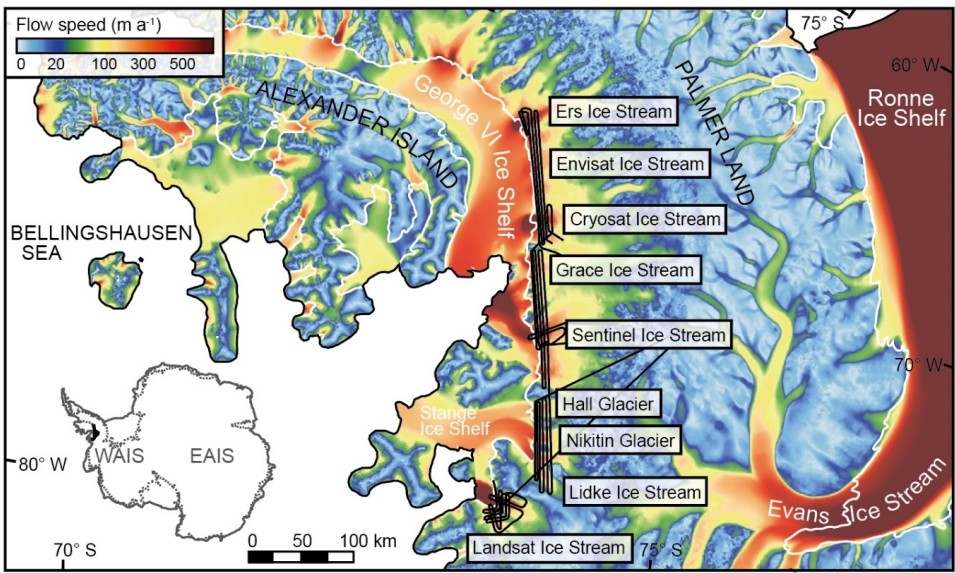

**Figure 1.** Airborne radio-echo sounding surveys (RES) (black lines), collected during the austral summer of 2016/2017, transcend the Bellingshausen Sea sector of Palmer Land in the Antarctic Peninsula. RES surveys transect several glaciers and ice streams along the English Coast at, or close to the Antarctic Surface Accumulation and Ice Discharge (ASAID) grounding line (white line) (Bindschadler et al., 2011), after which the ice floats. Background imagery shows surface flow speeds from MEaSUREs (Mouginot et al., 2019). The inset map shows the location of RES surveys used in this paper, superimposed on a map of Antarctica.

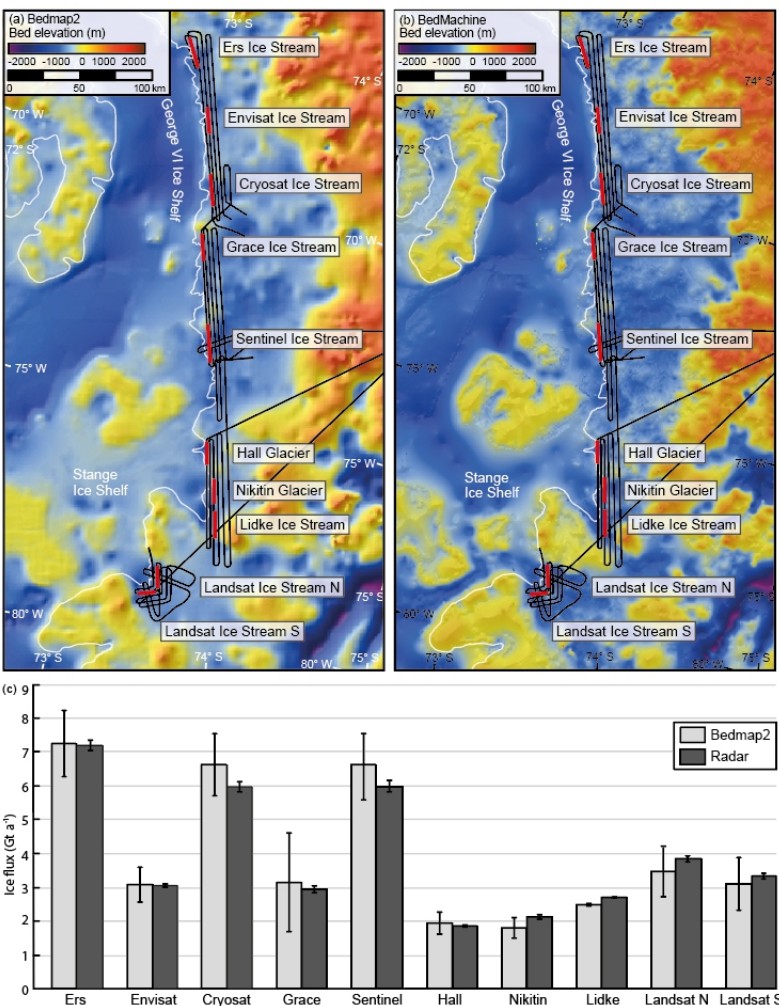

**Figure 2.** Major outlet glacier and ice stream flux gates (red) along the English Coast of Palmer Land. Subglacial topography maps from Bedmap2 (Fretwell et al., 2013) and BedMachine (Morlighem, 2019) are presented in panels (a) and (b). Black lines denote airborne RES transects detailed in this paper, whilst the white line shows the location of the ASAID grounding line (Bindschadler et al., 2011). Both maps show that subglacial topography frequently rests well below sea level along the English Coast. Panel (c) compares ice flux measurements (in gigatons), derived from Bedmap2 ice thickness data (Fretwell et al., 2013) (light grey bars) and direct radar measurements (dark grey bars), used as an input to BedMachine (Morlighem, 2019). These calculations utilise the same flux gates, noted in (a) and (b).

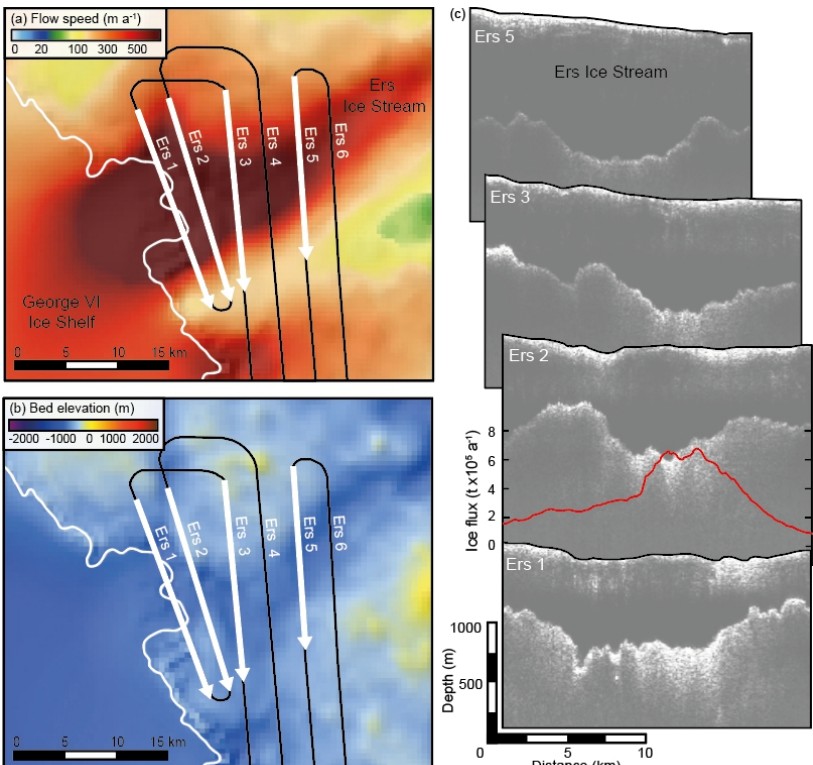



**Figure 3.** Ice penetrating radar transects (black lines) across Ers Glacier, superimposed on a map of surface flow
speeds (Mouginot et al., 2019) (a) and subglacial topography from BedMachine (Morlighem, 2019) (b). White
arrows indicate the location and direction of radargrams presented in (c) whilst the white line indicates the ASAID
grounding line (Bindschadler et al., 2011). Ice flux across RES transect Ers 2 is displayed as a red line in (c). Note
that the scale is in tonnes x$10^5$. c) Radargrams reveal surface topography, ice thickness and the subglacial bed
(recorded as diffuse white reflectors).

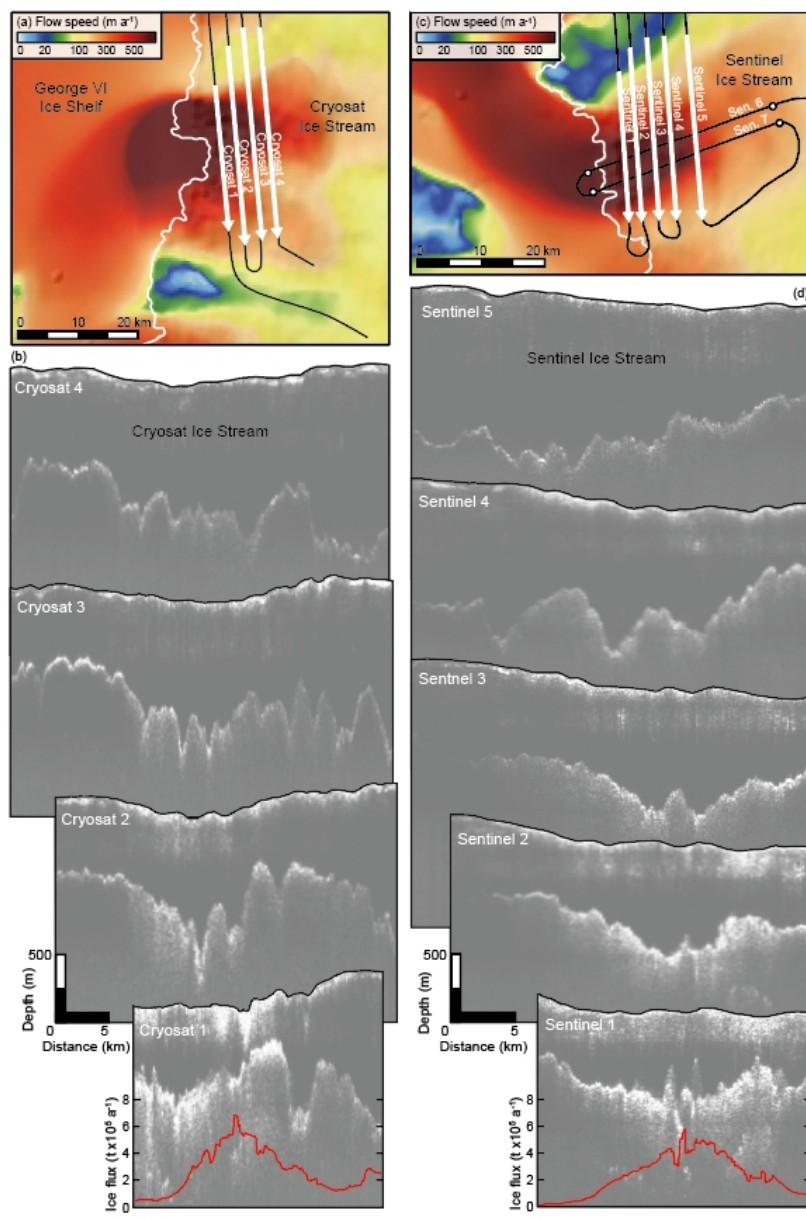



**Figure 4.** Radar investigations of Cryosat and Sentinel ice streams. Surface flow speed maps (Mouginot et al.,
2019) reveal the spatial variability in flow in panels (a) and (c). These panels highlight the location of radargrams
collected along the English Coast (black lines) as well as the direction and location of radargrams (white arrows)
displayed in (b) and (d). White lines indicate the ASAID grounding line (Bindschadler et al., 2011) whilst white
circles in (c) represent the extent of along-flow radar transects presented in Figure 7. Red lines in b) and d) show
calculated ice flux along RES transects Cryosat 1 and Sentinel 1.

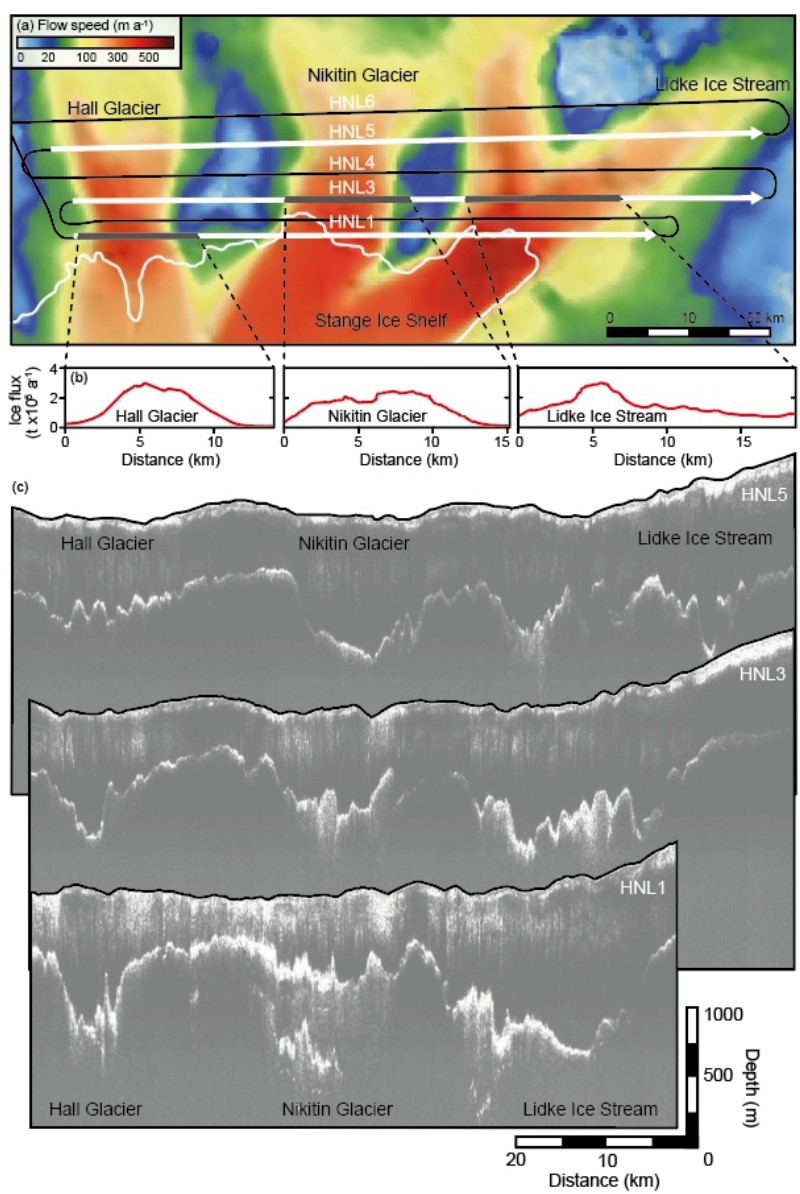



**Figure 5.** Hall Glacier, Nikitin Glacier and Lidke Ice Stream transfer fast flowing ice to the local grounding line
(white), where ice flow coalesces in the Stange Ice Shelf. (a) surface flow speeds from Mouiginot et al. (2019),
superimposed with English Coast radargram tracks (black), white arrows to indicate the location and direction of
radargrams presented in (c), and thick grey lines to note ice flux gates, graphed in (b). The white line indicates the
ASAID grounding line (Bindschadler et al., 2011). Note that the map has been rotated 90 degrees from its true
orientation (shown in Figure 1). Radargrams in (c) reveal changes in ice thickness and subglacial topography
down flow.

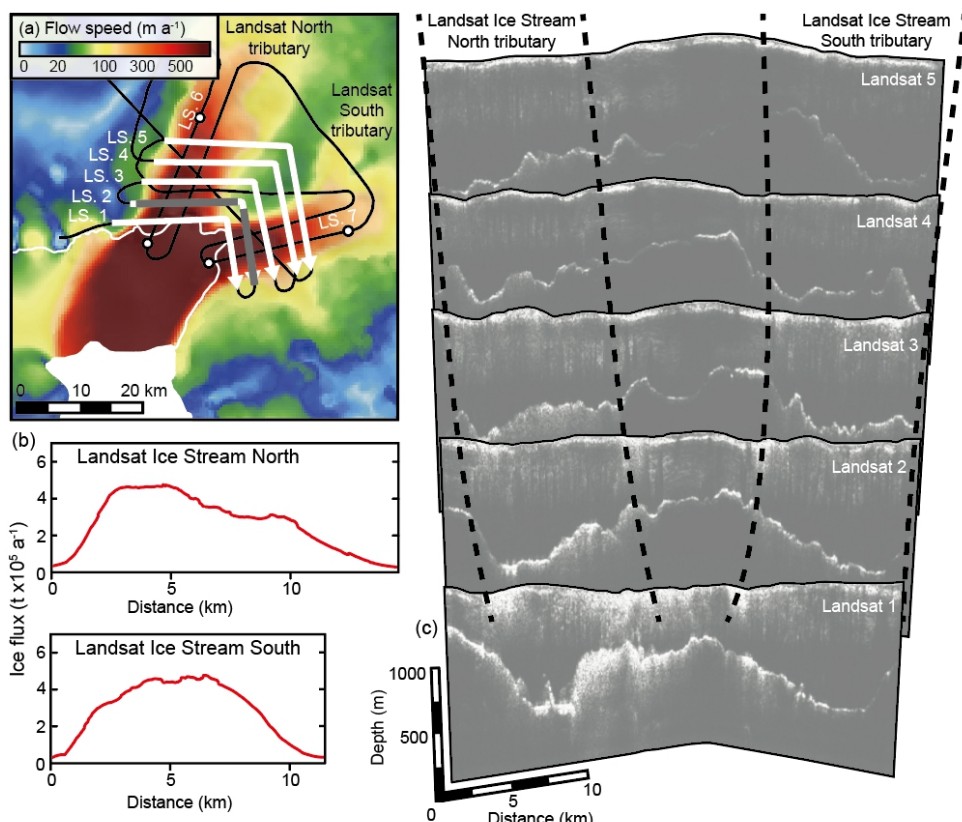



**Figure 6.** Landsat Ice Stream is fed by northern and southern tributaries which coalesce at the grounding zone.
These discrete flow units are clearly visible in (a) which shows a map of surface flow speeds from Mouginot et
al. (2019). Black lines show the density of RES transects in this location, whilst white arrows show the location
and orientation of transects displayed in (c). In panel (a) the white line represents the ASAID grounding line
(Bindschadler et al., 2011) whilst thick grey lines show the location of flux gates, presented in (b). White circles
on panel (a) represent the extent of along-flow radar transects presented in Figure 7. Radargrams in (c) show how
the two ice stream tributaries (approximately marked by a black dashed line) are separated by relatively high
elevation subglacial topography.

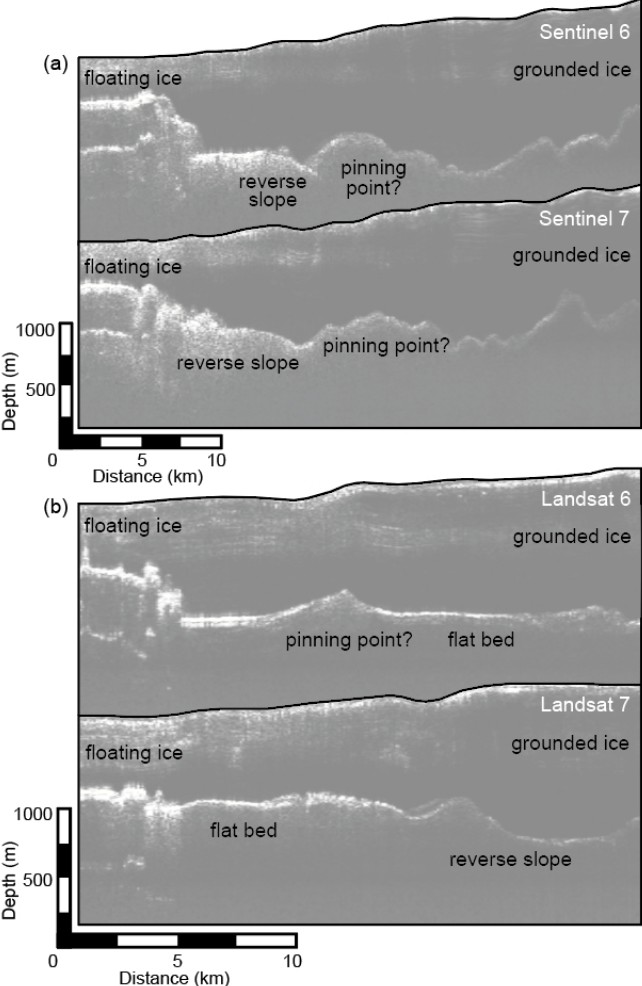

**Figure 7.** Along-flow radar transects of Sentinel Ice Stream (a) and Landsat Ice Stream (b). Transect locations
are marked by circles in Figures 4c and 6a. All four radargrams reveal a general pattern of surface lowering and
ice sheet thinning down flow (from right to left). Bright, white, diffuse reflectors on the left-hand side of the
radargrams represent floating ice and water ingress. Annotations highlight these features, and basal conditions.