# Peer review of "Subglacial topography and ice flux along the English Coast of Palmer Land, Antarctic Peninsula 2"

_Earth System Science Data, 2020_

## Referee Comment (RC1) · Joseph MacGregor (Referee) · 24 Aug 2020

Review of "Subglacial topography and ice flux along the English Coast of Palmer Land, Antarctic Peninsula" by K. Winter et al.

Joseph A. MacGregor 24 August 2020

This manuscript concerns a new airborne radar-sounding dataset collected across Palmer Land along the southwestern coastline of the Antarctic Peninsula. The manuscript reports the motivation for collecting the dataset, the instrument used, visuals of the data collected and their overall significance. In this case, the bed topography near the grounding zone of multiple outlet glaciers is described in substantial detail, with its potential consequences for the future evolution of these glaciers carefully con-

sidered.

This is a carefully structured manuscript on a new and important dataset for Antarctic science. Having reviewed several similar manuscripts in the past, this one is undoubtedly among the best of this category. Thorough, fair and thoughtful about the data collected. I found little to fault and consider my concerns no more than minor. The most significant of these can be grouped into two categories: 1. How ice flux is calculated and represented, even though it is not a direct part of the dataset reported it is discussed at some length. Some of these concerns may require recalculation of ice flux. 2. The other potential implications of the dataset.

Which dataset is shown/used in this study? The 1us or 4us pulse radargrams? I went to the DOI and found both. Should be stated in the MS in §3.1.

116: This value of the radio-wave velocity (168 m/us) is equivalent to a real part of the relative permittivity of 3.1844. CReSIS typically uses a value of 3.15 (double-check what they use for Antarctica, but I believe it's the same as for Greenland). Was the difference reconciled prior to the cross-over analysis?

121: Include appropriate data citation to MCoRDS data, e.g., was it this one? https://nsidc.org/data/IRMCR2/versions/1

124-125: It's understandable that high-elevation OIB transits where MCoRDS was operating would produce worse cross-over comparisons, particularly over regions of high topographic relief. However, as written it's a little unclear if Paxman et al. (2019) did the same cross-over analysis on this dataset but without the high-elevation flights? My broader recommendation is that high-elevation MCoRDS transits (assuming >2000 m AGL as opposed to typical ∼500 m AGL) not be included in the cross-over analysis even if they were included in BedMachine v1. Currently what's discussed is not just an apples-to-apples comparison between low-AGL PASIN2 and low-AGL MCoRDS.

153: Better than "close to" would be "immediately upstream of".

153: The history of flux-gate selection for these types of analyses is surprisingly complex relative to the task at hand (a line on a map) and involves various decisions that limited reproducibility of past landmark results. Based on the reported dataset itself, the authors' decision-making here is sensible, clearly explained and illustrated. However, I encourage the authors to consider adding a comparison against modern, openly distributed fluxgate positions and flux values, in particular Gardner et al. (2018; doi:10.5194/tc-12-521-2018). Not because their locations are better (they certainly aren't, given that they predate this dataset), but because they offer a direct point of comparison against the best recent study involving fluxgates in this region.

Further, I strongly encourage the authors to amend their excellent Figure 2 with a flux calculation from BedMachine v1 also. Why do this, given that these data were directly used in BedMachine? Because BedMachine's algorithm does not require perfect fidelity to radar observations, its uncertainty assessment is different and it is also much more likely to be the source of choice for independent flux calculations in the future.

161: The value of ice density used is oddly precise given the other given the other values used, sometimes with only one significant figure (e.g., 10 m firn correction). I understand that this is a reference value for ice density, but does that mean a value so precise is justified for this real-world application?

The discussion of ice thickness, flux and bed topography is excellent. It clearly outlines the similarities and differences between the outlet glaciers in this region. There is some limited discussion of apparent bed reflectivity that is fine, but no direct discussion of the potential value of whether this radar system/survey is likely to permit robust reflectivity analysis or the potential for analysis of radiostratigraphy (e.g., continuity index or direct tracing). A paragraph on these topics would be helpful to contextualize these datasets for obvious other applications. The radargrams focus on the bed and don't highlight any coherent radiostratigraphy, and I wouldn't expect much in this region to begin with, but still that should be clearly stated.
Figures

The figures are in general excellent and should be commended. The presentations of the radargrams, bed topography and flux are direct and clear.

All figures except 2# (bed elevation) use a rainbow color scale (surface speed), whose future is bleak. I recommend switching to another color scale for longevity, e.g., "hot" in MATLAB. Further, given the spatial scales considered, I'm not getting much out of the continuous color bar. I strongly recommend switching to a discrete one ($\leq$20 intervals).

I've never seen an L-shaped scale bar before, and it mostly works. However, much of the discussion concerning the topographic setting wisely revolves around the bed elevation not just ice thickness. It would be very helpful for the reader if the radargrams could be amended with a horizontal line at the elevation of sea level, so that it becomes unambiguous how deep the various troughs are.

Grammar, etc.

121: IceBridge 227: they are about three times slower 228: few tens of metres 308: use same unit of tonnes as the rest of the MS, i.e., Gt 403: order of hundreds of metres 435: principal investigator

---

## Referee Comment (RC2) · Anonymous Referee #2 · 28 Aug 2020

The manuscript by Winter et al. is a well written presentation of a new ice thickness data set covering nine ice streams and glaciers draining parts of Palmer Land on the Antarctic Peninsula into George VI Ice Shelf and Stange Ice Shelf, respectively directly into Bellingshausen Sea. The authors calculate the mass flux across the survey ice stream and glacier as show case for an application of the new ice thickness data set.

The chosen structure is clear and the used data are easily accessible as described. Several suggested minor changes are given below.

Paragraphs 3.1 and 3.2 provide only very brief information on the RES hardware and obtained data and need to be complemented. It would be helpful, if either in paragraph 3.2 or 7 the used RES profiles are listed.

[Figure]

The wave propagation speed in ice (0.168 m/ns, L168) is in contradiction to the ice density (916,7 kg/m3, L 161) used for calculation of the ice fluxes. Using the equation given by Kovacz et al., 1995, the propagation speed should be 0.1684 m/ns (916,7 kg/m3), respectively the density 923,3 kg/m3. Why was a density of 916.7 kg/m3 chosen?

In addition to the suggested adjustments above, the authors might consider to re-arrange paragraphs 4 and 5, so the structure of Results and Discussion is the same.

Suggested adjustments/corrections:

- L 14/15, etc. including figures: Please correct spelling of ERS, CryoSat, and GRACE Ice Stream. According to https://geonames.usgs.gov are these ice streams spelled in the same way as the satellites.

- L46, 48, 54, 74, 402, 405, 409, 410: Please replace IPR for RES as introduced in L12.

- L 102 ff: The data repository contains two data sets (1 and 4 microsecond). This fact and how they are recorded should be explained in this paragraph.

- L 107: Jeoffry et al. 2018 provide hardly more details on the deployed RES system (PSAIN-2) then given here. Corr et al. 2007 seems be a more suitable reference (cited by Jeoffry et al.). at least for an earlier version of PASIN2.

- L 114: Please provide version no and source for PROMAX (similar to ReflexW om L 129).

- L 121: Please provide reference for OIB data used.

- L 131: Please provide version no and reference to the free OpendTect package.

- L 135ff: Similar to L 102ff, please point out which of the two available data sets you are referring to.

- L 148, 276, 308, 315, 317, 326, 367, 369: Please replace ">" by suitable wording or exact value.

- L 151: Which data set in the repository contains the high-resolution ice thickness measurements?

- L 177: The argument, only the largest ice streams and glacier are presented in paragraphs 4 and 5 is irritating, respectively not correct, because the not presented Enivsat Ice Streeam and GRACE Ice Stream are not the smallest among the covered ice stream and glaciers, see Figure 2c.

- L 190, 210, 226, 246, 255, 276. 291: Please use same format for all given fluxes, preferably using two digits.

- L 227, 228, 403: Please replace numbers and "x" by words.

- L 487-489: Jeoffry et al, 2018: Pages and DOI need to be updated: p. 711-725 and https://doi.org/10.5194/essd-10-711-2018.

- Figure 1: Please improve readability of given names and labels, e.g. Stange Ice Shelf, and mark the investigated area in the overview inset. Please provide at least to labels/tics per side.

- Figures 3-7: It would be helpful to provide the exaggeration factors of the radargrams.

- All Figures: two clearly different colour scales would make it easier to distinguish between bed elevation and flow speed.
* * *

---

## Short Comment (SC1) · 28 Aug 2020

Many thanks for your thoughtful comments. We are delighted to receive such a positive response!

As you say, all comments and suggestions are no more than minor, so in this short, early response we'd like to focus on the tasks that may require a little more than a simple text-change:

We will re-run the cross-over analysis to explore the impact of high elevation OIB transits.

The complexity of flux gate selection is something we are very considerate of, so we are glad that you approve of our methods. As different flux gates will always yield

different flux values, we do not feel that a direct, numerical comparison to previously published flux gate values from different locations (such as Gardner et al., 2018) would be beneficial. We will however look into this in more detail to see if we can include additional information to the paper to address these comments.

The addition of a paragraph about radiostratigraphy is an excellent idea. We will add this in.

Thank you for your positive comments about our figures. We will explore your figure suggestions in due course.

Hopefully this short, early response will give you an indication of how we plan to proceed with paper revisions.

Thanks again,

Kate

---

## Short Comment (SC2) · 28 Aug 2020

Many thanks for your thoughtful comments.

As the suggested changes are minor we will address each suggestion in due course, amending the manuscript where necessary.

As a point of discussion, we would like to note that we use the Antarctic Place-names Committee approved spelling of ice streams and glaciers along the English Coast: https://apc.antarctica.ac.uk/gazetteers/latest-additions-bat/. It is therefore our intention to leave the spelling as it is in our original submission.

We hope you agree with our spelling decision and we look forward to incorporating your suggested changes in our revised manuscript.

[Figure]

Thanks again,

Kate

---

## Author Comment (AC1) · 20 Oct 2020

We thank both referees for constructive and supportive reviews. For referee 1 (Joseph MacGregor), we will add BedMachine values into Figure 2c and note sea level on each of the radar figures. We will also compare our findings to Gardner et al. 2018. These changes are all minor – but they will strengthen the paper.

For referee 2, there are very few changes required. We will explore the density value, as you suggest, and we will make the minor text-based changes to the manuscript where required. Following our previous response to your suggested changes, we note that we will keep the glacier names as they were in the original manuscript – as these are the names approved by the Antarctic Place-names Committee. In making these

changes, we feel the paper will be stronger, but with limited alteration.

We really appreciate the time you have both spent on reviewing our work.

Kate Winter (on behalf of co-authors)

———————————————————

---

## Author Response (AR1)

**Response to reviewer 1:**

*"This manuscript concerns a new airborne radar-sounding dataset collected across Palmer Land along the southwestern coastline of the Antarctic Peninsula. The manuscript reports the motivation for collecting the dataset, the instrument used, visuals of the data collected and their overall significance. In this case, the bed topography near the grounding zone of multiple outlet glaciers is described in substantial detail, with its potential consequences for the future evolution of these glaciers carefully considered."*

*"This is a carefully structured manuscript on a new and important dataset for Antarctic science. Having reviewed several similar manuscripts in the past, this one is undoubtedly among the best of this category. Thorough, fair and thoughtful about the data collected. I found little to fault and consider my concerns no more than minor."*

We would like to thank reviewer 1 for taking the time to provide a number of helpful comments and suggestions. We are delighted to receive such a positive review.

*Which dataset is shown/used in this study? The 1us or 4us pulse radargrams? I went to the DOI and found both. Should be stated in the MS in §3.1.*

We use the 1us radargrams. This has now been added to section 7 – with reference notes in section 3.1 and 3.4.

116: This value of the radio-wave velocity (168 m/us) is equivalent to a real part of the relative permittivity of 3.1844. CReSIS typically uses a value of 3.15 (double-check what they use for Antarctica, but I believe it's the same as for Greenland). Was the difference reconciled prior to the cross-over analysis?

This analysis was conducted by the aerogeophysical team at the British Antarctic Survey who provided us with the RES dataset.

*121: Include appropriate data citation to MCoRDS data, e.g., was it this one? https://nsidc.org/data/IRMCR2/versions/1*

We have now included this reference in the manuscript.

*124-125: It's understandable that high-elevation OIB transits where MCoRDS was operating would produce worse cross-over comparisons, particularly over regions of high topographic relief. However, as written it's a little unclear if Paxman et al. (2019) did the same cross-over analysis on this dataset but without the high-elevation flights? My broader recommendation is that high-elevation MCoRDS transits (assuming >2000 m AGL as opposed to typical _500 m AGL) not be included in the cross-over analysis even if they were included in BedMachine v1. Currently what's discussed is not just an apples-to-apples comparison between low-AGL PASIN2 and low-AGL MCoRDS.*

We have decided to remove the reference to Paxman et al. 2019 in section 3.1 as it confuses matters without any real benefit to the paper. We now exclude the high elevation flights from our crossover analysis too.

153: Better than "close to" would be "immediately upstream of".

Replaced.

*153: The history of flux-gate selection for these types of analyses is surprisingly complex relative to the task at hand (a line on a map) and involves various decisions that limited reproducibility of past landmark results. Based on the reported dataset itself, the authors' decision-making here is sensible, clearly explained and illustrated However, I encourage the authors to consider adding a comparison against modern, openly distributed fluxgate positions and flux values, in particular Gardner et al.*

*(2018; doi:10.5194/tc-12-521-2018). Not because their locations are better (they certainly aren't, given that they predate this dataset), but because they offer a direct point of comparison against the best recent study involving fluxgates in this region.*
As we did not reference the Gardner et al. 2018 paper in our initial submission we have now included a flux gate comparison from the paper in section 6. This new text is deliberately brief, because our flux gate analysis is very different (Gardner et al. 2018 use basin-wide flux estimates over a much larger flux gate area, and their basal topography and velocity datasets are obviously quite different to the ones we present) so it will never be an apples-to-apples comparison, as you point out too.

*Further, I strongly encourage the authors to amend their excellent Figure 2 with a flux calculation from BedMachine v1 also. Why do this, given that these data were directly used in BedMachine? Because BedMachine's algorithm does not require perfect fidelity to radar observations, its uncertainty assessment is different and it is also much more likely to be the source of choice for independent flux calculations in the future.*
We have now added BedMachine flux calculations to Figure 2 and provided commentary related to Figure 2c in Section 6.

*161: The value of ice density used is oddly precise given the other given the other values used, sometimes with only one significant figure (e.g., 10 m firn correction). I understand that this is a reference value for ice density, but does that mean a value so precise is justified for this real-world application?*
Agreed. We have recalculated our ice fluxes in response to your comment, and a similar comment from Reviewer 2. We now use 917 kg m$^{-3}$ because it is consistent with the value used in Bedmap2 and BedMachine. We have amended the text accordingly.

*The discussion of ice thickness, flux and bed topography is excellent. It clearly outlines the similarities and differences between the outlet glaciers in this region. There is some limited discussion of apparent bed reflectivity that is fine, but no direct discussion of the potential value of whether this radar system/survey is likely to permit robust reflectivity analysis or the potential for analysis of radiostratigraphy (e.g., continuity index or direct tracing). A paragraph on these topics would be helpful to contextualize these datasets for obvious other applications. The radargrams focus on the bed and don't highlight any coherent radiostratigraphy, and I wouldn't expect much in this region to begin with, but still that should be clearly stated.*
Agreed. We have now included this information in Section 7.

*The figures are in general excellent and should be commended. The presentations of the radargrams, bed topography and flux are direct and clear.*
*All figures except 2# (bed elevation) use a rainbow color scale (surface speed), whose future is bleak. I recommend switching to another color scale for longevity, e.g., "hot" in MATLAB. Further, given the spatial scales considered, I'm not getting much out of the continuous color bar. I strongly recommend switching to a discrete one (_20 intervals).*
Thanks for your kind comments about our figures. We tried out the hot colour scale you suggested (see below), and we have tried other colour scales in a similar theme but we feel these lack the detailed information we present with our original colour scale so we have left the colour scale as it was in the original manuscript.

[Figure]

[Figure]

*I've never seen an L-shaped scale bar before, and it mostly works. However, much of the discussion concerning the topographic setting wisely revolves around the bed elevation not just ice thickness. It would be very helpful for the reader if the radargrams could be amended with a horizontal line at the elevation of sea level, so that it becomes unambiguous how deep the various troughs are.*
We have now included sea level, or a reference to sea level in each of the radargram figures/captions. Thanks for this suggestion.

*121: IceBridge*
Changed.

*227: they are about three times slower*
Changed.

*228: few tens of metres*
Changed.

*308: use same unit of tonnes as the rest of the MS, i.e., Gt*
Changed.

*403: order of hundreds of metres*
Changed.

*435: principal investigator*
Changed.

**Response to reviewer 2:**

*"The manuscript by Winter et al. is a well written presentation of a new ice thickness data set covering nine ice streams and glaciers draining parts of Palmer Land on the Antarctic Peninsula into George VI Ice Shelf and Stange Ice Shelf, respectively directly into Bellingshausen Sea. The authors calculate the mass flux across the survey ice stream and glacier as show case for an application of the new ice thickness data set. The chosen structure is clear and the used data are easily accessible as described. Several suggested minor changes are given below."*

We would like to thank reviewer 2 for taking the time to provide a number of helpful comments and suggestions. We are grateful for their positive review.

*Paragraphs 3.1 and 3.2 provide only very brief information on the RES hardware and obtained data and need to be complemented. It would be helpful, if either in paragraph 3.2 or 7 the used RES profiles are listed.*

We have now added this information to section 7 and referred readers to this information in section 3.1

*The wave propagation speed in ice (0.168 m/ns, L168) is in contradiction to the ice density (916,7 kg/m3, L 161) used for calculation of the ice fluxes. Using the equation given by Kovacz et al., 1995, the propagation speed should be 0.1684 m/ns (916,7 kg/m3), respectively the density 923,3 kg/m3. Why was a density of 916.7 kg/m3 chosen?*

Thank you for highlighting this discrepancy in ice density to us. We have explored the impact of density on flux outputs in the figure below. A density of 923.3kg m$^{-3}$ makes very limited difference to the flux outputs we present in the paper (less than 0.7% overall). Given that using a fixed value of ice density across each flux gate is already a simplification (which is ubiquitous to Antarctic geophysics), we do not believe that small differences in density value can introduce errors in our results. Following comments from Reviewer 1, we realise that our original reference value for ice density of 916.7 kg m$^{-3}$ was perhaps too precise. Instead, for consistency and ease of comparison, we have chosen to use the ice density value of 917 kg m$^{-3}$, which is utilised in both Bedmap2 and BedMachine - which are the main compilations used in continent wide flux gate and numerical modelling studies. We have now made this clear in the text presented in section 3.4.

[Figure]

*In addition to the suggested adjustments above, the authors might consider to rearrange paragraphs 4 and 5, so the structure of Results and Discussion is the same.*

We are unsure what you mean here. In the comments above you have made reference to sections 3.1 and 3.2 as paragraphs. With that line of thought it looks like you want us to change round the results and discussion sections. That wouldn't make much sense to us.

*Suggested adjustments/corrections:*
*- L 14/15, etc. including figures: Please correct spelling of ERS, CryoSat, and GRACE Ice Stream. According to https://geonames.usgs.gov are these ice streams spelled in the same way as the satellites.*
As noted in our early online discussion - we use the Antarctic Place-names Committee approved spelling of ice streams and glaciers along the English Coast: https://apc.antarctica.ac.uk/gazetteers/latest-additions-bat/. We will therefore keep the spelling as it is in our original submission.

*- L46, 48, 54, 74, 402, 405, 409, 410: Please replace IPR for RES as introduced in L12.*
Replaced.

*- L 102 ff: The data repository contains two data sets (1 and 4 microsecond). This fact and how they are recorded should be explained in this paragraph.*
Apologies for this oversight. We have added this information to Section 7 and referred to this information in sections 3.1 and 3.4.

*- L 107: Jeoffry et al. 2018 provide hardly more details on the deployed RES system (PSAIN-2) then given here. Corr et al. 2007 seems be a more suitable reference (cited by Jeoffry et al.). at least for an earlier version of PASIN2.*
We have replaced the Jeoffry et al. (2018) reference with Corr et al. (2007) in Section 3.1 as per your suggestion.

*- L 114: Please provide version no and source for PROMAX (similar to ReflexW om L 129).*
This has now been added to the manuscript in section 3.1.

*- L 121: Please provide reference for OIB data used.*
Reference now included.

*- L 131: Please provide version no and reference to the free OpendTect package.*
Version number and reference to software provider now included.

*- L 135ff: Similar to L 102ff, please point out which of the two available data sets you are referring to.*
We have now noted this information in various locations throughout the manuscript (in response to your comment on line 102).

*- L 148, 276, 308, 315, 317, 326, 367, 369: Please replace ">" by suitable wording or exact value.*
The > symbol has now been removed from the manuscript.

*- L 151: Which data set in the repository contains the high-resolution ice thickness measurements?*
New text added to Section 3.4 to address this comment – referring the reader to section 7 where we have included this information.

*- L 177: The argument, only the largest ice streams and glacier are presented in paragraphs 4 and 5 is irritating, respectively not correct, because the not presented Enivsat Ice Streeam and GRACE Ice Stream are not the smallest among the covered ice stream and glaciers, see Figure 2c.*
Apologies. We have now amended this statement.

*- L 190, 210, 226, 246, 255, 276. 291: Please use same format for all given fluxes, preferably using two digits.*
Flux values now have a consistent format, with two decimal places.

*- L 227, 228, 403: Please replace numbers and "x" by words.*
Replaced.

*- L 487-489: Jeoffry et al, 2018: Pages and DOI need to be updated: p. 711-725 and https://doi.org/10.5194/essd-10-711-2018.*
This reference has been removed from the manuscript in response to your earlier suggestion.

*- Figure 1: Please improve readability of given names and labels, e.g. Stange Ice Shelf, and mark the investigated area in the overview inset. Please provide at least to labels/tics per side.*
We have made these labels larger and semi-bold so they now stand out more. The investigated area was marked on the overview inset, but we have now changed the outline colour of Antarctica to make this more obvious. There are two tics per side to show lat/long but the top one is hidden a little by the flow speed key. We have used the tics to show direction, so 70 degrees south can be traced up if the reader so-wishes.

*- Figures 3-7: It would be helpful to provide the exaggeration factors of the radargrams.*
We have given this suggestion some thought but we have decided not to include this in the figures as we feel that this information is already available in the scale bar.

*- All Figures: two clearly different colour scales would make it easier to distinguish between bed elevation and flow speed.*
Thanks for your suggestion. The only time we show both plots together is in Figure 3 – where the colour scale clearly shows warm colours indicating fast flow in panel (a) and cold colours where there is low elevation subglacial topography in panel (b). We have carefully considered your suggestion but ultimately decided that we are happy with our original colour scale choice.